# Learning to Schedule Heuristics in Branch and Bound

**Antonia Chmiela**
Zuse Institute Berlin, Germany
`chmiela@zib.de`

**Elias B. Khalil**
University of Toronto, Canada
`khalil@mie.utoronto.ca`

**Ambros Gleixner**
Zuse Institute Berlin, Germany
HTW Berlin, Germany
`gleixner@zib.de`

**Andrea Lodi**
CERC, Polytechnique Montréal, Canada
`andrea.lodi@polymtl.ca`

**Sebastian Pokutta**
Zuse Institute Berlin, Germany
Technische Universität Berlin, Germany
`pokutta@zib.de`

## Abstract

Primal heuristics play a crucial role in exact solvers for Mixed Integer Programming
(MIP). While solvers are guaranteed to find optimal solutions given sufficient time,
real-world applications typically require finding good solutions early on in the
search to enable fast decision-making. While much of MIP research focuses
on designing effective heuristics, the question of how to manage multiple MIP
heuristics in a solver has not received equal attention. Generally, solvers follow
hard-coded rules derived from empirical testing on broad sets of instances. Since
the performance of heuristics is problem-dependent, using these general rules for a
particular problem might not yield the best performance. In this work, we propose
the first data-driven framework for scheduling heuristics in an exact MIP solver.
By learning from data describing the performance of primal heuristics, we obtain
a problem-specific schedule of heuristics that collectively find many solutions at
minimal cost. We formalize the learning task and propose an efficient algorithm for
computing such a schedule. Compared to the default settings of a state-of-the-art
academic MIP solver, we are able to reduce the average primal integral by up to
$49\%$ on two classes of challenging instances.

## 1 Introduction

Many decision-making problems arising from real-world applications can be formulated using *Mixed
Integer Programming (MIP)*. The *Branch and Bound* (B&B) framework is a general approach to
solving MIPs to global optimality. Over the recent years, the idea of using machine learning (ML)
to improve optimization techniques has gained renewed interest. There exist various approaches to
tackle different aspects of the solving process using classical ML techniques. For instance, ML has
been used to find good parameter configurations for a solver (Hutter et al., 2009, 2011), improve
node (He et al., 2014), variable (Khalil et al., 2016; Gasse et al., 2019; Nair et al., 2020) or cut
(Baltean-Lugojan et al., 2019) selection strategies, and detect decomposable structures (Kruber et al.,
2017).

Even though exact MIP solvers aim for global optimality, finding good feasible solutions fast is at
least as important, especially in the presence of a time limit. The use of *primal heuristics* is crucial to

35th Conference on Neural Information Processing Systems (NeurIPS 2021).

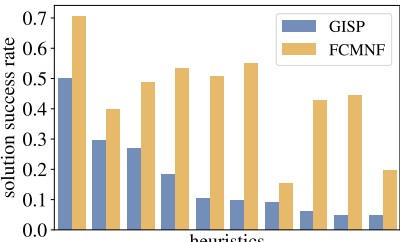

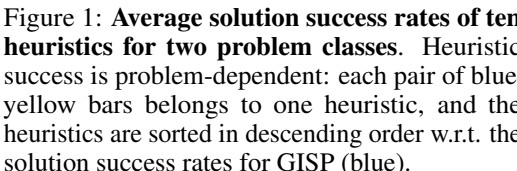

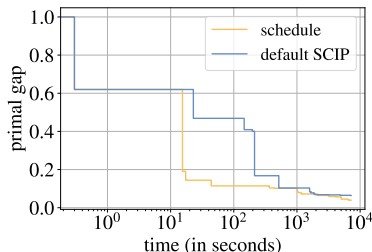

Figure 1: **Average solution success rates of ten heuristics for two problem classes**. Heuristic success is problem-dependent: each pair of blue-yellow bars belongs to one heuristic, and the heuristics are sorted in descending order w.r.t. the solution success rates for GISP (blue).

Figure 2: **Primal gap for an exemplary GISP instance.** Our method's heuristic schedule (orange) obtains better solutions earlier than SCIP's default (blue).

ensuring good primal performance in modern solvers. For instance, Berthold (2013a) showed that the primal bound–the objective value of the best solution–improved on average by around $80\%$ when primal heuristics were used. Generally, a solver includes a variety of primal heuristics, where each class of heuristics (e.g., rounding, diving, large-neighborhood search) exploits a different idea to find good solutions. During B&B, some of these heuristics are executed successively at each node of the search tree, and improved solutions, if any, are reported back to the solver. An extensive overview of different primal heuristics, their computational costs, and their impact in MIP solving can be found in Lodi (2013); Berthold (2013b, 2018).

Since most heuristics can be very costly, it is necessary to be strategic about the order in which the heuristics are executed and the number of iterations allocated to each. Such decisions are often made by following hard-coded rules derived from testing on broad benchmark test sets. While these static rules yield good performance on average, their performance can be far from satisfactory when considering specific families of instances. To illustrate this fact, Figure 1 compares the solution success rates, i.e., the fraction of calls to a heuristic where a solution was found, of different primal heuristics for two problem classes: the *Generalized Independent Set Problem (GISP)* (Hochbaum and Pathria, 1997; Colombi et al., 2017) and the *Fixed-Charge Multicommodity Network Flow Problem (FCMNF)* (Hewitt et al., 2010).

In this paper, we propose a data-driven approach to systematically improve the use of primal heuristics in B&B. By learning from data about the duration and success of every heuristic call for a set of training instances, we construct a *schedule of heuristics* that specifies the ordering and duration for which each heuristic should be executed to obtain good primal solutions early on. As a result, we are able to significantly improve the use of primal heuristics as shown in Figure 2 for one MIP instance.

**Contributions.** Our main contributions can be summarized as follows:

1. **We formalize the learning task** of finding an effective, cost-efficient heuristic schedule on a training dataset as a Mixed Integer Quadratic Program (Section 3);

2. We propose an **efficient heuristic** for solving the training (scheduling) problem and a **scalable data collection** strategy (Sections 4 and 5);

3. We perform **extensive computational experiments** on a class of challenging instances and **demonstrate the benefits of our approach** (Section 6).

**Related Work.** Optimizing the use of primal heuristics is a topic of ongoing research. For instance, by characterizing nodes with different features, Khalil et al. (2017) propose an ML method to decide when to execute heuristics to improve primal performance. After that decision, all heuristics are executed according to the predefined rules set by the solver. Hendel (2018) and Hendel et al. (2018) use bandit algorithms for the online learning of a heuristic ordering. The method proposed in this paper jointly adapts the ordering and duration for which each heuristic runs. Primal performance can also be improved using algorithm configuration (Hutter et al., 2009, 2011), a technique which is generally computational expensive since it relies on many black-box evaluations of the solver as its parameter configurations are evaluated and does not exploit detailed information about the effect

of parameter values on performance, e.g., how parameters of primal heuristics affect their success rates. There has also been work done on how to schedule algorithms optimally. Kadioglu et al. (2011) solved the problem for a portfolio of different MIP solvers whereas Hoos et al. (2014) focused on Answer Set Programming. Furthermore, Seipp et al. (2015) propose an algorithm that greedily finds a schedule of different parameter configurations for automated planning.

## 2 Preliminaries

Let us consider a MIP of the form

$$\min_{x \in \mathbb{R}^n} c^\mathsf{T} x \ \text{ s.t. } \ Ax \leq b, \ x_i \in \mathbb{Z}, \forall i \in I, \tag{$P_{MIP}$}$$

with matrix $A \in \mathbb{R}^{m \times n}$, vectors $c \in \mathbb{R}^n$, $b \in \mathbb{R}^m$, and a non-empty index set $I \subseteq [n]$ for integer variables. A MIP can be solved using B&B, a tree search algorithm that finds an optimal solution to ($P_{MIP}$) by recursively partitioning the original problem into linear subproblems. The nodes in the resulting search tree correspond to these subproblems. Throughout this work, we assume that each node has a unique index that identifies the node even across B&B trees obtained for different MIP instances. For a set of instances $\mathcal{X}$, we denote the union of the corresponding node indices by $\mathcal{N}_\mathcal{X}$.

**Primal Performance Metrics.** Since we are interested in finding good solutions fast, we consider a collection of different metrics for primal performance. Beside statistics like the time to the first/best solution and the solution/incumbent success rate, we mainly focus on the *primal integral* (Berthold, 2013a) as a comprehensive measure of primal performance. Intuitively, this metric can be interpreted as a normalized average of the incumbent value over time. A formal definition can be found in Appendix A. Figure 2 gives an example for the primal gap function. The primal integrals are the areas under each of the curves. It is easy to see that finding near-optimal incumbents earlier shrinks the area under the graph of the primal gap, resulting in a smaller primal integral.

## 3 Data-Driven Heuristic Scheduling

Since the performance of heuristics is highly problem-dependent, it is natural to consider *data-driven* approaches for optimizing the use of primal heuristics for the instances of interest. Concretely, we consider the following practically relevant setting. We are given a set of heuristics $\mathcal{H}$ and a homogeneous set of training instances $\mathcal{X}$ from the same problem class. In a data collection phase, we are allowed to execute the B&B algorithm on the training instances, observing how each heuristic performs at each node of each search tree. At a high level, our goal is to then leverage this data to obtain a schedule of heuristics that minimizes a primal performance metric.

The specifics of how such data collection is carried out will be discussed later on in the paper. First, let us examine the decisions that could potentially benefit from a data-driven approach. Our discussion is inspired by an in-depth analysis of how the open-source MIP solver SCIP (Gamrath et al., 2020) manages primal heuristics. However, our approach is generic and is likely to apply to other solvers.

**Controlling the Order.** One important degree of freedom in scheduling heuristics is the order in which a set of heuristics $\mathcal{H}$ is executed by the solver at a given node. This can be controlled by assigning a *priority* for each heuristic. In a *heuristic loop*, the solver then iterates over the heuristics in decreasing priority. The loop is terminated if a heuristic finds a new incumbent solution that cuts off the current node. As such, an ordering $\langle h_1, \ldots, h_k \rangle$ that prioritizes effective heuristics can lead to time savings without sacrificing primal performance.

**Controlling the Duration.** Furthermore, solvers use working limits to control the computational effort spent on heuristics. Consider diving heuristics as an example. Increasing the maximal diving depth increases the likelihood of finding an integer feasible solution. At the same time, this increases the overall running time. Figure 3 visualizes this cost-benefit trade-off empirically for three different diving heuristics, highlighting the need for a careful "balancing act". For a heuristic $h \in \mathcal{H}$, let $\tau \in \mathbb{R}_{>0}$ denote its time budget. Then, we are interested in finding a *schedule* $S := \langle (h_1, \tau_1), \ldots, (h_k, \tau_k) \rangle, h_i \in \mathcal{H}$. Since controlling the time budget directly can be unreliable and lead to nondeterministic behavior in practice (see Appendix E for details), a deterministic proxy measure is preferable. For diving heuristics, the maximal diving depth provides a suitable measure as demonstrated by Figure 3. Similar measures can be used for other types of heuristics, as we will

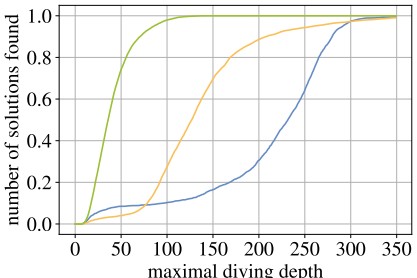 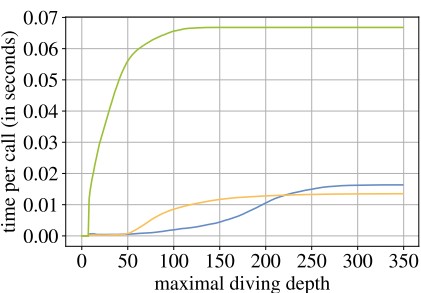

Figure 3: **Number of solutions found and cost of different diving heuristics depending on the the maximal diving depth**: This figure shows the average fraction of solutions found by a heuristic (left) and average duration in seconds (right) of three diving heuristics when limiting the maximal depth of a dive. Hereby, the baseline for the values on the vertical axis of the left figure is the number of solutions found by the heuristics with no limitations on the diving depth. The likelihood of finding a solution increases with the maximal diving depth. At the same time, an average call to all three heuristics becomes more expensive as the diving depth increases.

demonstrate with Large Neighborhood Search heuristics in Section 6. In general, we will refer to $\tau_i$ as the maximal number of *iterations* that is allocated to a heuristic $h_i$ in schedule $S$.

**Deriving the Scheduling Problem.** Having argued for order and duration as suitable control decisions, we will now formalize our heuristic scheduling problem. Ideally, we would like to construct a schedule $S$ that minimizes the primal integral, averaged over the training set $\mathcal{X}$. Unfortunately, it is very difficult to optimize the primal integral directly, as it depends on the *sequence* of incumbents found over time during B&B. It also depends on the way the search tree is explored, which is affected by pruning, further complicating any attempt at directly optimizing this primal metric.

We address this difficulty by considering a more tractable surrogate objective. Recall that $\mathcal{N}_\mathcal{X}$ denotes the collection of search tree nodes of the set of training instances $\mathcal{X}$. We will construct a schedule $S$ that finds feasible solutions for a large fraction of the nodes in $\mathcal{N}_\mathcal{X}$, while also minimizing the number of iterations expended by schedule $S$. Note that we consider feasible solutions instead of incumbents here: this way, we are able to obtain more data faster since a heuristic finds a feasible solution more often than a new incumbent. The framework we propose in the following can handle incumbents instead, but we have found no benefit in doing so in preliminary experiments.

For a heuristic $h$ and node $N$, denote by $t(h, N)$ the iterations necessary for $h$ to find a solution at node $N$, and set $t(h, N) = \infty$ if $h$ does not succeed at $N$. Now suppose a schedule $S$ is successful at node $N$, i.e., some heuristic finds a solution within the budget allocated to it in $S$. Let $j_S = \min\{j \in [|\mathcal{H}|] : t(h_j, N) \leq \tau_j\}$ be the index of the first successful heuristic. Following the execution of $h_{j_S}$, the heuristic loop is terminated, and the time spent by $S$ at node $N$ is given by

$$T(S, N) := \sum_{i \in [j_S - 1]} \tau_i + t(h_{j_S}, N).$$

Otherwise, set $T(S, N) := \sum_{i=1}^{k} \tau_i + 1$, where the additional 1 penalizes unsolved nodes.

Furthermore, let $\mathcal{N}_S$ denote the set of nodes at which schedule $S$ is successful in finding a solution. Then, we consider the heuristic scheduling problem given by

$$\min_{S \in \mathcal{S}} \sum_{N \in \mathcal{N}_\mathcal{X}} T(S, N) \text{ s.t. } |\mathcal{N}_S| \geq \alpha |\mathcal{N}_\mathcal{X}|. \qquad (P_S)$$

Here $\alpha \in [0, 1]$ denotes the minimum fraction of nodes for which the schedule must find a feasible solution. Problem $(P_S)$ can be formulated as a Mixed-Integer Quadratic Program (MIQP); the complete formulation can be found in Appendix B.

To find such a schedule, we need to know $t(h, N)$ for every heuristic $h$ and node $N$. Hence, when collecting data for the instances in the training set $\mathcal{X}$, we track for every B&B node $N$ at which a heuristic $h$ was called, the number of iterations $\tau_N^h$ it took $h$ to find a feasible solution; we set $\tau_N^h = \infty$ if $h$ does not succeed at $N$. Formally, we require a training dataset $\mathcal{D} := \big\{ (h, N, \tau_N^h) \mid$

$h \in \mathcal{H}, N \in \mathcal{N}_{\mathcal{X}}, \tau_N^h \in \mathbb{R}_{>0} \cup \{\infty\}\}$. Section 5 describes a computationally efficient approach for building $\mathcal{D}$ using a *single* B&B run per training instance.

## 4   Solving the Scheduling Problem

Problem $(P_{\mathcal{S}})$ is a generalization of the Pipelined Set Cover Problem which is known to be $\mathcal{NP}$-hard (Munagala et al., 2005). As for the MIQP in Appendix B, tackling it using a non-linear integer programming solver is challenging: the MIQP has $O(|\mathcal{H}||\mathcal{N}_{\mathcal{X}}|)$ variables and constraints. Since a single instance may involve thousands of search tree nodes, this leads to an MIQP with hundreds of thousands of variables and constraints even with a handful of heuristics and tens of training instances.

As mentioned in Related Work, algorithm configuration tools such as SMAC (Hutter et al., 2011) could be used to solve $(P_{\mathcal{S}})$ heuristically. Since SMAC is a sequential algorithm that searches for a good parameter configuration by successively adapting and re-evaluating its best configurations, its running time can be quite substantial. In the following, we present a more efficient approach.

We now direct our attention towards designing an efficient heuristic algorithm for $(P_{\mathcal{S}})$. A similar problem was studied by Streeter (2007) in the context of decision problems. Among other things, the author discusses how to find a schedule of (randomized) heuristics that minimizes the expected time necessary to solve a set of training instances $\mathcal{X}$ of a decision problem. Although this setting is somewhat similar to ours, there exist multiple aspects in which they differ significantly:

1. *Decision problems are considered instead of MIPs:* Solving a MIP is generally much different from solving a decision problem. When using B&B, we normally have to solve many linear subproblems. Since in theory, every such LP is an opportunity for a heuristic to find a new incumbent, we consider the set of nodes $\mathcal{N}_{\mathcal{X}}$ instead of $\mathcal{X}$ as the "instances" we want to solve.

2. *A heuristic call can be suspended and resumed:* In the work of Streeter, a heuristic can be executed in a "suspend-and-resume model": If $h$ was executed before, the action $(h, \tau)$ represents *continuing* a heuristic run for an additional $\tau$ iterations. When $h$ reaches the iteration limit, the run is suspended and its state kept in memory such that it can be resumed later in the schedule. This model is not used in MIP solving due to challenges in maintaining the states of heuristics in memory. As such, we allow every heuristic to be included in the schedule at most once.

3. *Time is used to control the duration of a heuristic run:* Controlling time directly is unreliable in practice and can lead to nondeterministic behavior of the solver. Instead, we rely on different proxy measures for different classes of heuristics. Thus, when building a schedule that contains heuristics of distinct types, we need to ensure that these measures are comparable.

Despite these differences, it is useful to examine the greedy scheduling approach proposed in Streeter (2007). A schedule $G$ is built by successively adding the action $(h, \tau)$ that maximizes the ratio of the marginal increase in the number of instances solved to the cost (i.e., $\tau$) of including $(h, \tau)$. As shown in Corollary 2 of Streeter (2007), the *greedy schedule $G$* yields a 4-approximation to that version of the scheduling problem. In an attempt to leverage this elegant heuristic in our problem $(P_{\mathcal{S}})$, we will describe it formally.

Let us denote the greedy schedule by $G := \langle g_1, \ldots, g_k \rangle$. Then, $G$ is defined inductively by setting $G_0 = \langle \rangle$ and $G_j = \langle g_1, \ldots, g_j \rangle$ with

$$g_j = \operatorname*{argmax}_{(h,\tau) \in \mathcal{H}_{j-1} \times \mathcal{T}} \frac{|\{N \in \mathcal{N}_{\mathcal{X}}^{j-1} \mid \tau_N^h \leq \tau\}|}{\tau}.$$

Here, $\mathcal{H}_j$ denotes the set of heuristics that are not in $G_j$, $\mathcal{N}_{\mathcal{X}}^j$ denotes the subset of nodes not solved by $G_j$, and $\mathcal{T}$ is the interval generated by all possible iteration limits in $\mathcal{D}$, i.e., $\mathcal{T} := [\min\{\tau_N^h \mid (N, h, \tau_N^h) \in \mathcal{D}\}, \max\{\tau_N^h \mid (N, h, \tau_N^h) \in \mathcal{D}\}]$. We stop adding actions $g_j$ when $G_j$ finds a solution at all nodes in $\mathcal{N}_{\mathcal{X}}$ or all heuristics are contained in the schedule, i.e., $\mathcal{H}_j = \emptyset$.

Unfortunately, the resulting schedule can perform arbitrarily bad in our setting: Assume we have $|\mathcal{N}_{\mathcal{X}}| = 100$ and only one heuristic $h$. This heuristic solves one node in just one iteration and requires 100 iterations for each of the other 99 nodes. Following the greedy approach, the resulting schedule would be $G = \langle (h, 1) \rangle$ since $\frac{1}{1} > \frac{99}{100}$. Whenever $\alpha > 0.01$, $G$ would be infeasible for our

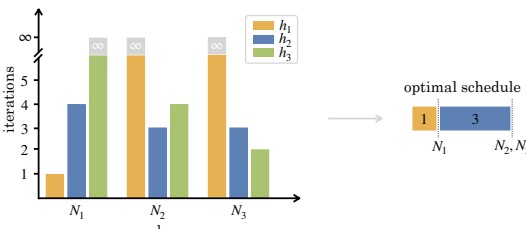

Figure 4: **Illustration of a toy dataset and the greedy schedule:** The data is shown on the left and the (optimal) schedule obtained by following the greedy algorithm is illustrated on the right.

constrained problem ($P_\mathcal{S}$). Since we are not allowed to add a heuristic more than once, this cannot be fixed with the current algorithm.

To avoid this situation, we propose the following modification. Instead of only considering the heuristics that are not in $G_{j-1}$ when choosing the next action $g_j$, we also consider the option to run the last heuristic $h_{j-1}$ of $G_{j-1}$ for longer. That is, we allow to choose $(h_{j-1}, \tau)$ with $\tau > \tau_{j-1}$. Note that the cost of adding $(h_{j-1}, \tau)$ to the schedule is not $\tau$, but $\tau - \tau_{j-1}$, since we decide to run $h_{j-1}$ for $\tau - \tau_{j-1}$ iterations longer and not to rerun $h_{j-1}$ for $\tau$ iterations.

Furthermore, when including different classes of heuristics in the schedule, the respective time measures are not necessarily comparable. We observed that not taking the difference of iteration cost into account led to an increase of the primal integral of up to 23% compared to default SCIP. To circumvent this problem, we use the average time per iteration to normalize different notions of iterations. We denote the average cost of an iteration by $t_{avg}^h$ for heuristic $h$. Note that $t_{avg}^h$ can be easily computed by tracking the running time of a heuristic during data collection. Hence, we redefine $g_j$ and obtain

$$g_j = \operatorname*{argmax}_{(h,\tau)\in\mathcal{A}_{j-1}} \frac{|\{N \in \mathcal{N}_\mathcal{X}^{j-1} \mid \tau_N^h \leq \tau\}|}{c_{j-1}(h,\tau)},$$

with $\mathcal{A}_j := (\mathcal{H}_j \times \mathcal{T}) \cup \{(h_j, \tau) \mid \tau > \tau_j, \tau \in \mathcal{T}\}$ and

$$c_j(h,\tau) := \begin{cases} t_{avg}^h \tau, & \text{if } h \neq h_j \\ t_{avg}^h(\tau - \tau_j), & \text{otherwise.} \end{cases}$$

We set $\mathcal{A}_0 := \mathcal{H} \times \mathcal{T}$ and $c_0(h,\tau) = t_{avg}^h \tau$. With this modification, we obtain the schedule $G = \langle(h, 100)\rangle$ (which solves all 100 nodes) in the above example.

Additionally, it is also possible to consider the quality of the found solutions when choosing the next action $g_j$. Since we observed that the resulting schedules increased the primal integral by up to 11%, we omit this here.

Finally, note that this greedy procedure still does not explicitly enforce that the schedule is successful at a fraction of at least $\alpha$ nodes. In our experiments, however, we observe that the resulting schedules reach a success rate of $\alpha = 98\%$ or above. The final algorithm can be found in Appendix C.

**Example.** Figure 4 shows an example of how we obtain a schedule with three heuristics and nodes. As indicated by the left figure, the dataset is given by

$$\mathcal{D} = \{(h_1, N_1, 1), (h_1, N_2, \infty), (h_1, N_3, \infty), (h_2, N_1, 4), (h_2, N_2, 3), (h_2, N_3, 3), (h_3, N_1, \infty), (h_3, N_2, 4), (h_3, N_3, 2)\}.$$

Let us now assume that all three heuristic have the same costs, i.e., $t_{avg}^{h_1} = t_{avg}^{h_2} = t_{avg}^{h_3}$. We build the schedule $G$ as follows. First, we add action $(h_1, 1)$, since $h_1$ solves one node with only one iteration, yielding the best ratio. Since $N_1$ is "solved" by the current schedule and $h_1$ cannot solve any other nodes, both $N_1$ and $h_1$ do not need to be considered anymore. Among the remaining possibilities, the action $(h_2, 3)$ is the best, since $h_2$ solves both nodes in three iterations yielding a ratio of $\frac{2}{3}$. In contrast, executing $h_3$ for two and four iterations, respectively, yields a ratio of $\frac{1}{2}$. Hence, we add $(h_2, 3)$ to $G$ and obtain $G = \langle(h_1, 1), (h_2, 3)\rangle$. The schedule then solves all three nodes as shown on the right of Figure 4. Note that this schedule is an optimal solution of ($P_\mathcal{S}$) for $\alpha > \frac{1}{3}$.

# 5 Data Collection

The scheduling approach described thus far rests on the availability of a dataset $\mathcal{D}$. Among others, each entry in $\mathcal{D}$ stores the number of iterations $\tau_N^h$ required by heuristic $h$ to find a feasible solution at node $N$. This piece of information must be collected by executing the heuristic and observing its performance. Two main challenges arise in collecting such a dataset for multiple heuristics:

1. *Efficient data collection:* Solving MIPs by B&B remains computationally expensive, even given the sophisticated techniques implemented in today's solvers. This poses difficulties to ML approaches that create a single reward signal per MIP evaluation, which may take several minutes up to hours. In other words, even with a handful of heuristics, i.e., a small set $\mathcal{H}$, it is prohibitive to run B&B once for each heuristic-training instance pair in order to construct the dataset $\mathcal{D}$.

2. *Obtaining unbiased data:* Executing multiple heuristics at each node of the search tree during data collection can have dangerous side effects: if a heuristic finds an incumbent, subsequent heuristics are no longer executed at the same node, as described in Section 3.

We address the first point by using a specially-crafted version of the MIP solver for collecting *multiple reward signals* for the execution of *multiple heuristics* per single MIP evaluation during the training phase. As a result, we obtain a large amount of data points that scales with the running time of the MIP solves. This has the clear advantage that the efficiency of our data collection does not automatically decrease when the time to evaluate a single MIP increases for more challenging problems.

To prevent bias from mutual interaction of different heuristics during training, we engineered the MIP solver to be executed in a special *shadow mode*, where heuristics are called in a sandbox environment and interaction with the main solving path is maximally reduced. In particular, this means that new incumbents and primal bounds are not communicated back, but only recorded for training data. This setting is an improved version of the shadow mode introduced in Khalil et al. (2017).

As a result of these measures, we have instrumented the SCIP solver in a way that allows for the collection of a proper dataset $\mathcal{D}$ with a *single run* of the B&B algorithm per training instance.

# 6 Computational Results

The code we use for data collection and scheduling is publicly available.[1]

## 6.1 Heuristics and Instances

We can build a schedule containing arbitrary heuristics as long as there is a time measure available. We focus on two broad groups of complex heuristics: *Diving* and *Large Neighborhood Search (LNS)*. Both classes are much more computationally expensive than simpler heuristics such as rounding (for which scheduling is not necessary and executions are extremely fast), but are generally also more likely to find (good) solutions (Berthold, 2006). That is why it is particularly important to schedule these heuristics most economically.

**Diving Heuristics.** Diving heuristics examine a single probing path by successively fixing variables according to a specific rule. There are multiple ways of controlling the duration of a dive. After careful consideration, we decided on using the maximum diving depth to limit the cost of a call to a diving heuristic: It is both related to the effort spent by the heuristic and its likelihood of success.

**LNS Heuristics.** These heuristics first build a neighborhood of some reference point which is then searched for improving solutions by solving a sub-MIP. To control the duration, we choose to limit the number of nodes in the sub-MIP. The idea behind this measure is similar to limiting the diving depth of diving heuristics: In both cases, we control the number of subproblems a heuristic considers within its execution. Nevertheless, the two measures are not directly comparable: The most expensive LNS heuristic was on average around 892 times more expensive than the cheapest diving heuristic.

To summarize, we schedule 16 primal heuristics: ten diving and six LNS heuristics. By controlling this set, we cover about $\frac{2}{3}$ of the more complex heuristics implemented in SCIP. The remaining heuristics are executed after the schedule according to their default settings.

---

[1] `https://github.com/antoniach/heuristic-scheduling`

We focus on two problem classes which are challenging on the primal side: The *Generalized Independent Set Problem (GISP)* (Hochbaum and Pathria, 1997; Colombi et al., 2017) and the *Fixed Charge Multicommodity Network Flow Problem (FCMNF)* (Hewitt et al., 2010). For GISP, we generate two types of instances: The first one takes graphs from the 1993 DIMACS Challange which is also used by Khalil et al. (2017) and Colombi et al. (2017) (120 for training and testing) and the second type uses randomly generated graphs as a base (25 for training and 10 for testing). The latter is also used to obtain FCMNF instances (20 for training and 120 for testing). A detailed description of the problems and how we generate and partition the instances can be found in Appendix D.

## 6.2 Results

To study the performance of our approach, we used the state-of-the-art solver SCIP 7.0 (Gamrath et al., 2020) with CPLEX 12.10.0.0 as the underlying LP solver. Thereby, we needed to modify SCIP's source code to collect data as described in Section 5, as well as control heuristic parameters that are not already implemented by default. For our experiments, we used a Linux cluster of Intel Xeon CPU E5-2660 v3 2.60GHz with 25MB cache and 128GB main memory. The time limit in all experiments was set to two hours; for data collection to four hours. Because the primal integral depends on time, we ran one process at a time on every machine, allowing for accurate time measurements. Furthermore, since MIP solver performance can be highly sensitive to even small and seemingly performance-neutral perturbations during the solving process (Lodi and Tramontani, 2013), we implemented an exhaustive testing framework that uses four random seeds and evaluates schedules trained with one data distribution on other data distributions, a form of transfer learning.

The main baseline we compare against is default SCIP. Note that since the adaptive diving and LNS methods presented in Hendel (2018) and Hendel et al. (2018) are included in default SCIP as heuristics, we implicitly compare to these methods when comparing to default SCIP; improvements due to our method reflect improvements over Hendel's approach. Furthermore, we also consider SCIP_TUNED, a hand-tuned version of SCIP's default settings for GISP.[2] Since in practice, a MIP expert would try to manually optimize some parameters when dealing with a homogeneous set of instances, we emulated that process to create an even stronger baseline to compare against.

**GISP – Random graph instances.** Table 1 (rows DIVING) shows partial results of the transfer learning experiments for schedules with diving heuristics (see Table 3 in Appendix F for the complete table). Our scheduling framework yields a significant improvement w.r.t. primal integral on all test sets. Since this improvement is consistent over all schedules and test sets, we are able to confirm that the behavior actually comes from our procedure. Especially remarkable is the fact that the schedules trained on smaller instances also perform well on much larger instances. Furthermore, we can see that the schedules perform especially well on instances of increasing difficulty (size). This behavior is intuitive: Since our method aims to improve the primal performance of a solver, there is more room for improvement when an instance is more challenging on the primal side. Over all test sets, the schedules terminated with a strictly better primal integral on 69–76% and with a strictly better primal bound on 59–70% of the instances compared to SCIP_TUNED (see Table 4 in Appendix F for details). In addition, the number of incumbents found by the heuristics considered in the schedule increased significantly: 49–61% of the incumbents were found by heuristics in the schedule, compared to only 33% when running with default SCIP (see Table 4 in Appendix F for details).

Table 1 (rows DIVING+LNS) shows the transfer learning experiments for schedules containing diving and LNS heuristics. By including both types of heuristics, we are able to improve over the diving-only schedule in around half of the cases, since on the instances we consider, diving seems to perform significantly better than LNS. Furthermore, we also observe less consistent performance among the schedules which leads us to the conclusion that LNS's behavior is harder to predict. How to further improve our scheduling procedure to better fit LNS is part of future work.

**GISP – Finding a schedule with SMAC.** As mentioned earlier, we can also find a schedule by using the algorithm configuration tool SMAC. To test SMAC's performance on the random graph instances, we trained ten SMAC schedules, each with a different random seed, on each of the five training sets. We used the primal integral as a performance metric. To make it easier for SMAC, we only considered diving heuristics. We gave SMAC the same total computational time for training as we did in data collection: With 25 training instances per set using a four hour time limit each, this comes

---

[2]We set the frequency offset to 0 for all diving heuristics.

| train \ test | schedule | [150,160] | [250,260] | [350,360] | [450,460] | [550,560] |
|---|---|---|---|---|---|---|
| [150,160] | DIVING | $0.89 \pm 0.23$ | $0.87 \pm 0.37$ | $0.87 \pm 0.28$ | $0.78 \pm 0.24$ | $0.65 \pm 0.24$ |
| | DIVING+LNS | $0.87 \pm 0.26$ | $0.76 \pm 0.29$ | $0.85 \pm 0.22$ | $0.95 \pm 0.20$ | $0.82 \pm 0.26$ |
| | SMAC | $0.81 \pm 0.23$ | $0.77 \pm 0.34$ | $0.90 \pm 0.27$ | $0.85 \pm 0.24$ | $0.65 \pm 0.19$ |
| [250,260] | DIVING | $0.89 \pm 0.28$ | $0.81 \pm 0.34$ | $0.92 \pm 0.23$ | $0.81 \pm 0.24$ | $0.66 \pm 0.20$ |
| | DIVING+LNS | $0.86 \pm 0.20$ | $0.77 \pm 0.27$ | $0.85 \pm 0.20$ | $0.85 \pm 0.18$ | $0.64 \pm 0.18$ |
| | SMAC | $0.87 \pm 0.26$ | $0.88 \pm 0.42$ | $0.87 \pm 0.25$ | $0.83 \pm 0.24$ | $0.59 \pm 0.22$ |
| [350,360] | DIVING | $0.84 \pm 0.24$ | $0.82 \pm 0.36$ | $0.81 \pm 0.26$ | $0.80 \pm 0.21$ | $0.59 \pm 0.20$ |
| | DIVING+LNS | $0.86 \pm 0.27$ | $0.85 \pm 0.29$ | $0.94 \pm 0.31$ | $0.99 \pm 0.28$ | $0.65 \pm 0.23$ |
| | SMAC | $0.86 \pm 0.24$ | $0.80 \pm 0.37$ | $0.86 \pm 0.25$ | $0.80 \pm 0.24$ | $0.68 \pm 0.18$ |
| [450,460] | DIVING | $0.89 \pm 0.25$ | $0.83 \pm 0.36$ | $0.77 \pm 0.23$ | $0.81 \pm 0.21$ | $0.58 \pm 0.20$ |
| | DIVING+LNS | $0.84 \pm 0.27$ | $0.76 \pm 0.31$ | $0.92 \pm 0.23$ | $0.86 \pm 0.24$ | $0.62 \pm 0.22$ |
| | SMAC | $0.93 \pm 0.26$ | $0.87 \pm 0.32$ | $0.90 \pm 0.19$ | $0.85 \pm 0.25$ | $0.69 \pm 0.23$ |
| [550,560] | DIVING | $0.88 \pm 0.26$ | $0.89 \pm 0.42$ | $0.86 \pm 0.27$ | $0.81 \pm 0.20$ | $0.63 \pm 0.21$ |
| | DIVING+LNS | $0.84 \pm 0.25$ | $0.78 \pm 0.35$ | $0.96 \pm 0.20$ | $0.88 \pm 0.27$ | $0.60 \pm 0.20$ |
| | SMAC | $0.87 \pm 0.22$ | $0.83 \pm 0.31$ | $0.92 \pm 0.29$ | $0.84 \pm 0.26$ | $0.58 \pm 0.21$ |
| SCIP_TUNED | - | $0.89 \pm 0.28$ | $0.99 \pm 0.31$ | $1.05 \pm 0.28$ | $0.94 \pm 0.23$ | $0.76 \pm 0.25$ |

Table 1: Average relative primal integral (mean $\pm$ std.) of schedule w.r.t. default SCIP over GISP instances derived from random graphs. The first fifteen rows correspond to schedules trained on instances of size [L,U] with different methods: DIVING (greedy schedule with diving), DIVING+LNS (greedy schedules with diving and LNS) and SMAC (SMAC-trained schedules with diving).

to 100 hours per training set and schedule. Note that since SMAC runs sequentially, training the SMAC schedules took over four days per schedule, whereas training a schedule following the greedy algorithm only took four hours with enough machines. To pick the best performing SMAC schedule for each training set, we ran all ten schedules on the test set of same size as the corresponding training set and chose the best performing one.

The results can be found in Table 1 (rows SMAC). As we can see, on all test sets, all schedules are significantly better than default SCIP. However, when comparing these results to the performance of the greedy schedules, we can see that SMAC performs worse on average. Over all five test sets, the SMAC schedules terminated with a strictly better primal integral on 36–54% and with a strictly better primal bound on 37–55% of the instances compared to its greedy counterparts.

**GISP – DIMACS graph instances.** The first three columns of Table 2 summarize the results on the instances derived from DIMACS graphs. As we can see, the schedule setting dominates default SCIP in all metrics, but an especially drastic improvement can be obtained w.r.t. the primal integral: the schedule reduces the primal integral by 49%. Furthermore, 92% of instances terminated with a strictly better primal integral and 57% with a strictly better primal bound. Even though SCIP_TUNED finds the best incumbent faster than the schedule, the latter terminates with a better primal bound (GISP is a maximization problem) explaining the small increase in time. When looking at the total time spent in heuristics, we see that heuristics run significantly shorter but with more success: On average, the incumbent success rate is higher compared to default SCIP. That the learned schedule not only improves the primal side of the problem, but also translates to an overall better performance is shown by the last two rows: SCHEDULE significantly dominates DEFAULT in the gap at termination as well as the primal-dual integral.

Compared to the results of the method in Khalil et al. (2017), where node features were used to decide if a heuristic should be executed, our scheduling procedure yields competitive performance: On average, their method reduced both the primal integral and the time to best incumbent by 60% (our method: 49% and 47%). Hereby it is important to note that our baseline (SCIP 7.0) is much faster than theirs (SCIP 3.2): for DIMACS instances, default SCIP terminated with a gap of 201.95% in Khalil et al. (2017) compared to 144.59% in our experiments. Furthermore, SCIP's technical reports show that version 7.0 is 58% faster than version 3.2 on a standard benchmark test set.

**FCMNF.** The last three columns of Table 2 summarize the results on the FCMNF instances. Also for this problem, we can see that the schedule setting dominates both DEFAULT and SCIP_TUNED in almost all metrics. In particular, we are able to almost double the number of solutions found and triple the incumbent success rate. Even though the improvement in the primal integral is not as drastic as we observed with GISP, it is still consistent over the whole test set: 62% of the instances

|  | GISP | | | FCMNF | | |
|---|---|---|---|---|---|---|
|  | DEFAULT | SCIP_TUNED | SCHEDULE | DEFAULT | SCIP_TUNED | SCHEDULE |
| Primal Integral | 934.48 | 555.75 | 470.73 | 618.52 | 608.19 | 564.07 |
| Time to first Incumbent | 1.33 | 1.33 | 1.26 | 34.89 | 34.08 | 27.36 |
| Time to best Incumbent | 4266.68 | 2642.46 | 2803.38 | 2973.66 | 3943.66 | 3782.45 |
| Best Incumbent | 2382.03 | 2385.73 | 2404.63 | 1242839 | 1227125 | 1221969 |
| Total heuristic calls* | 138.57 | 137.38 | 140.03 | 15.08 | 18.83 | 23.58 |
| Total heuristic time* | 258.88 | 304.96 | 190.10 | 1681.39 | 1620.02 | 1482.33 |
| Number of Incumbents found* | 2.72 | 3.08 | 3.33 | 0.14 | 0.25 | 0.23 |
| Incumbent Success Rate* | 0.01 | 0.02 | 0.02 | 0.01 | 0.03 | 0.02 |
| Gap | 144.59 | 144.03 | 141.70 | 9.64 | 8.19 | 7.85 |
| Primal-dual Integral | 450148.72 | 435321.67 | 430882.04 | 107894.01 | 107381.64 | 102855.88 |

Table 2: Summary of results on GISP instances derived from DIMACS graphs and for FCMNF instances (with schedules of diving heuristics). Values shown are aggregates over instances; geometric means are used. Statistics with * refer only to the heuristics used in the schedule.

terminated with a strictly better primal integral and 92% with a strictly better primal bound. Similar to the GISP results, SCHEDULE needs more time than DEFAULT to find the best incumbent, since it again terminates with a better primal bound (FCMNF is a minimization problem).

Finally, it is important to note that the trained schedules differ significantly from SCIP's default settings for all training sets. The improvements we observed when using these schedules supports our starting hypothesis, namely that the way default MIP solver parameters are set does not yield the best performance when considering specific use cases.

## 7    Conclusion and Discussion

In this work, we propose a data-driven framework for scheduling primal heuristics in a MIP solver such that the primal performance is optimized. Central to our approach is a novel formulation of the learning task as a scheduling problem, an efficient data collection procedure, and a fast, effective heuristic for solving the learning problem on a training dataset. A comprehensive experimental evaluation shows that our approach consistently learns heuristic schedules with better primal performance than SCIP's default settings. Furthermore, by replacing our heuristic algorithm with the algorithm configuration tool SMAC in our scheduling framework, we are able to obtain a worse but still significant performance improvement w.r.t. SCIP's default. Together with the prohibitive computational costs of SMAC, we conclude that for our heuristic scheduling problem, the proposed heuristic algorithm constitutes an efficient alternative to existing methods.

A possible limitation of our approach is that it produces a single, "one-size-fits-all" schedule for a class of training instances. It is thus natural to wonder whether alternative formulations of the learning problem leveraging additional contextual data about an input MIP instance and/or a heuristic can be useful. We note that learning a mapping from the space of MIP instances to the space of possible schedules is not trivial. The latter is a highly structured output space that involves both the permutation over heuristics and their respective iteration limits. The approach proposed here is much simpler in nature, which makes it easy to implement and incorporate into a sophisticated MIP solver.

Although we have framed the heuristic scheduling problem in ML terms, we are yet to analyze the learning-theoretic aspects of the problem. More specifically, our approach is justified on empirical grounds in Section 6, but we are yet to attempt to analyze potential generalization guarantees. We view the recent foundational results by Balcan et al. (2019) as a promising framework that may apply to our setting, as it has been used for the branching problem in MIP (Balcan et al., 2018).

## Disclosure of Funding

This work was partially funded by the Deutsche Forschungsgemeinschaft (DFG) under Germany's Excellence Strategy – The Berlin Mathematics Research Center MATH+ (EXC-2046/1, project ID: 390685689), and by the German Federal Ministry of Education and Research (BMBF) within the Research Campus MODAL (grant numbers 05M14ZAM, 05M20ZBM). Elias B. Khalil acknowledges support from the Scale AI Research Chair Program and an IVADO Postdoctoral Scholarship.

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
