# A  Primal Integral

If $x$ is feasible and $x^*$ is an optimal (or best known) solution to $(P_{MIP})$, the *primal gap* of $x$ is defined as

$$\gamma(x) := \begin{cases} 0, & \text{if } |c^\mathsf{T} x| = |c^\mathsf{T} x^*|, \\ 1, & \text{if } c^\mathsf{T} x \cdot c^\mathsf{T} x^* < 0, \\ \dfrac{|c^\mathsf{T} x - c^\mathsf{T} x^*|}{\max\{|c^\mathsf{T} x|, |c^\mathsf{T} x^*|\}}, & \text{otherwise.} \end{cases}$$

With $x^t$ denoting the incumbent at time $t$, the *primal gap function* $p : \mathbb{R}_{\geq 0} \to [0, 1]$ is then defined as

$$p(t) := \begin{cases} 1, & \text{if no incumbent is found until time } t, \\ \gamma(x^t), & \text{otherwise.} \end{cases}$$

For a time limit $T \in \mathbb{R}_{\geq 0}$, the primal integral $P(T)$ is then given by the area underneath the primal gap function $p$ up to time $T$, i.e., $P(T) := \sum_{i=1}^{K} p(t_{i-1})(t_i - t_{i-1})$, where $K - 1$ incumbents have been found until time $T$, and $t_0 = 0$, $t_K = T$, and $t_1, \ldots, t_{K-1}$ are the points in time at which new incumbents are found.

# B  Formulating the Scheduling Problem as a MIQP

In this section, we present the exact formulation of Problem $(P_S)$ as a MIQP. First, we describe the parameters and variables we need to formulate the problem. Then, we state the problem and explain shortly what every constraint represents.

## B.1  Parameters

- $\mathcal{D}$, set of data points coming from a set of MIP instances $\mathcal{X}$. Each data point is of the form $(h, N, \tau_N^h)$, where $h \in \mathcal{H}$ is a heuristic, $N \in \mathcal{N}_\mathcal{X}$ indexes a node of the B&B tree of $\mathcal{X}$, and $\tau_N^h$ is the number of iterations $h$ needed to find a solution to $N$ (if $h$ could not solve $N$, we set $\tau_N^h = \infty$).

- $T^h, \forall h \in \mathcal{H}$, the maximum number of iterations $h$ needed to find a solution, i.e., $T^h := \max\{\tau_N^h < \infty \mid N \in \mathcal{N}_\mathcal{X}\}$.

- $\alpha \in [0, 1]$, minimal fraction of nodes the resulting schedule should solve.

## B.2  Variables Describing the Schedule

- $x_p^h \in \{0, 1\}, \forall h \in \mathcal{H}, \forall p \in \{0, \ldots, |\mathcal{H}|\}$: The variable is set to 1 if $h$ is executed at position $p$ in the schedule (if $x_0^h = 1$ then $h$ is not in the schedule).

- $t^h \in [0, \ldots, T^h], \forall h \in \mathcal{H}$: This integer variable is equal to the maximal number of iterations $h$ can use in the schedule (we set $t^h = 0$ if $h$ is not in the schedule).

## B.3  Auxiliary Variables

- $p^h \in \{0, \ldots, |\mathcal{H}|\}, \forall h \in \mathcal{H}$: This integer variable is equal to the position of $h$ in the schedule.

- $s_N^h \in \{0, 1\}, \forall h \in \mathcal{H}, \forall N \in \mathcal{N}_\mathcal{X}$: The variable is set to 1 if heuristic $h$ solves node $N$ in the schedule.

- $s_N \in \{0, 1\}, \forall N \in \mathcal{N}_\mathcal{X}$: This variable is set to 1 if the schedule solves node $N$.

- $p_N^{min} \in \{1, \ldots, |\mathcal{H}|\}, \forall N \in \mathcal{N}_\mathcal{X}$: This integer variable is equal to the position of the heuristic that first solves node $N$ in the schedule (if the schedule does not solve $N$, we set it to $|\mathcal{H}|$).

- $z_N^h \in \{0, 1\}, \forall h \in \mathcal{H}, \forall N \in \mathcal{N}_\mathcal{X}$: This variable is set to 1 if $h$ is executed before position $p_N^{min}$.

- $f_N^h \in \{0, 1\}, \forall h \in \mathcal{H}, \forall N \in \mathcal{N}_{\mathcal{X}}$: The variable is set to 1 if $h$ is the heuristic that solves $N$ first, i.e., if $p^h = p_N^{min}$.

- $t_N \in \{1, \ldots, 1 + \sum_h T^h\}, \forall N \in \mathcal{N}_{\mathcal{X}}$: This integer variable is equal to the total number of iterations the schedule needs to solve node $N$ (if $N$ is not solved by the schedule, we set it to 1 plus the total length of the schedule, i.e., $1 + \sum_p \sum_h x_p^h t^h$).

## B.4 Formulation

In the following, we give an explicit formulation of $(P_{\mathcal{S}})$ as a MIQP. Note that some constraints use nonlinear functions like the maximum/minimum of a finite set or the indicator function $\mathbb{1}$. These can be easily linearized by introducing additional variables and constraints, thus the following formulation is indeed a MIQP. For the sake of readability, we omit stating all the linearizations explicitly.

$$\min \quad \sum_N t_N \tag{1}$$

$$\text{s.t.} \quad \sum_h x_p^h \leq 1, \forall p \in \{1, \ldots, |\mathcal{H}|\} \tag{2}$$

$$\sum_p x_p^h = 1, \forall h \in \mathcal{H} \tag{3}$$

$$p^h = \sum_p p x_p^h, \forall h \in \mathcal{H} \tag{4}$$

$$T^h(1 - x_0^h) \geq t^h, \forall h \in \mathcal{H} \tag{5}$$

$$s_N^h = \max\{0, \min\{1, t^h - \tau_N^h + 1\}\}, \forall h \in \mathcal{H}, \forall N \in \mathcal{N}_{\mathcal{X}} \tag{6}$$

$$s_N = \min\{1, \sum_h s_N^h\}, \forall N \in \mathcal{N}_{\mathcal{X}} \tag{7}$$

$$\frac{1}{|\mathcal{N}_{\mathcal{X}}|} \sum_N s_N \geq \alpha \tag{8}$$

$$p_N^{min} = \min\{p^h s_N^h + (1 - s_N^h) |\mathcal{H}|) \mid h \in \mathcal{H}\}, \forall N \in \mathcal{N}_{\mathcal{X}} \tag{9}$$

$$z_N^h = \mathbb{1}_{\{p^h < p_N^{min}\}}, \forall h \in \mathcal{H}, \forall N \in \mathcal{N}_{\mathcal{X}} \tag{10}$$

$$f_N^h = \mathbb{1}_{\{p^h = p_N^{min}\}}, \forall h \in \mathcal{H}, \forall N \in \mathcal{N}_{\mathcal{X}} \tag{11}$$

$$t_N = s_N(\sum_h z_N^h t^h + f_N^h \tau_N^h) + (1 - s_n)(1 + \sum_{h,p} x_p^h t^h), \forall N \in \mathcal{N}_{\mathcal{X}} \tag{12}$$

(1) calculates the total number of iterations the schedule needs to solve all nodes.

(2) and (3) guarantee that only one copy of each heuristic is run, and that every non-zero position is occupied by at most one heuristic.

(4) calculates the position of a heuristic in the schedule.

(5) ensures that $t^h = 0$ if $h$ is not in the schedule.

(6) forces $s_N^h$ to 1 if $h$ solves node $N$ in the schedule.

(7) forces $s_N$ to 1 if the schedule solves node $N$.

(8) guarantees that the schedules solves enough nodes.

(9) calculates the position of the first heuristic that solves $N$ in the schedule.

(10) forces $z_N^h$ to 1 if $h$ is executed before position $p_N^{min}$.

(11) forces $f_N^h$ to 1 if $h$ is executed at position $p_N^{min}$.

(12) calculates the number of iterations necessary for the schedule to solve $N$.

## C Pseudocode

In the following, we propose the greedy algorithm presented in Section 4 to obtain a heuristic schedule.

---
**Algorithm 1** Greedy algorithm to obtain a schedule

---
**Input:** Nodes $\mathcal{N}_\mathcal{X}$, heuristics $\mathcal{H}$, data $D$, time frame $\mathcal{T}$
**Output:** Greedy Schedule $G$
$G \leftarrow \langle \rangle$
$\mathcal{N}_{unsol} \leftarrow \mathcal{N}_\mathcal{X}$
$improve \leftarrow \text{TRUE}$
**repeat**
$\quad (h^*, \tau^*) \leftarrow \underset{(h,\tau)\in\mathcal{A}}{\operatorname{argmax}} \left[ \frac{|\{N\in\mathcal{N}_{unsol}|\tau_h^N \leq \tau\}|}{c(h,\tau)} \right]$
$\quad$ **if** $\frac{|\{N\in\mathcal{N}_{unsol}|\tau_{h^*}^N \leq \tau^*\}|}{c(h,\tau^*)} > 0$ **then**
$\quad\quad G \leftarrow G \oplus \langle (h^*, \tau^*) \rangle$
$\quad\quad \mathcal{N}_{unsol} \leftarrow \mathcal{N}_{unsol} \setminus \{N \in \mathcal{N}_{unsol} \mid \tau_{h^*}^N \leq \tau^*\}$
$\quad$ **else**
$\quad\quad improve \leftarrow \text{FALSE}$
$\quad$ **end if**
**until** $improve == \text{FALSE}$

---

## D Instances

In this section, we give a brief description of the two problems we consider in Section 6.2 and the partition we use for our experiments.

### D.1 Generalized independent set problem (GISP)

Let $G = (V, E)$ be a graph and $E' \subseteq E$ a subset of removable edges. Each vertex has a revenue and every edge has a cost associated with it. Then, GISP asks to select a subset of vertices and removable edges that maximizes the profit, i.e., the difference of vertex revenue and edge costs. Thereby, no edge should exist between two selected vertices $v, u \in V$, i.e., either we have that $(v, u) \notin E$ or $(v, u) \in E'$ is removed.

We generate GISP instances in the following way. Given a graph, we randomize the set of removable edges by setting the probability that an edge is in $E'$ to $\alpha = 0.75$. Furthermore, we choose the revenue for each node to be 100 and the cost of every edge as 1. This results in a configuration for which it is difficult to find good feasible solutions as shown in Colombi et al. (2017).

We use this scheme to generate two types of instances. The first one takes graphs from the 1993 DIMACS Challenge which is also used by Khalil et al. (2017); Colombi et al. (2017). Thereby, we focus on the same twelve dense graphs as well as the same train/test partition as in Khalil et al. (2017). We generate 20 instances for every graph by using different random seeds, leaving us with 120 instances for training as well as testing. For the second group of GISP instances, we use randomly generated graphs where the number of nodes is uniformly chosen from $\{L, \ldots, U\}$ for bounds $L, U \in \mathbb{N}$. An edge is added to the resulting graph with probability $\bar{\alpha} = 0.1$, giving us slightly less dense graphs than the previous case. We denote these sets by [L,U]. For each set, we generate 25 instances for training and 10 for testing.

### D.2 Fixed-Charge Multi-Commodity Flow Problem (FCMNF)

We are given a directed graph $G = (V, A)$ and commodities $k \in K$ such that a quantity of each $k$ needs to be routed from a source node to a destination node. Each arc $a \in A$ is characterized by a fixed cost that is only imposed when the arc is used, a maximal capacity, and a variable cost that is proportional to the flow traversing $a$. Then, FCMNF asks for a minimal cost routing of every commodity while respecting the arc capacities.

We generate FCMNF instances in the following way. Given a graph, we have 200 commodities with random source/destination nodes. We choose the commodity quantities to be uniformly distributed over $[10, 100]$, the fixed cost over $[1100, 5000]$, the capacity over $[10, 20000]$ and the variable costs over $[11, 50]$. To generate graphs, we again use the same randomized scheme as for GISP. For training, we generate 20 instances with $\{40, \ldots, 50\}$ nodes and for testing 120 instances with $\{75, \ldots, 125\}$ nodes.

## E   Implementation Details

Not every idea that works in theory can be directly translated to also work in practice. Hence, it is sometimes inevitable to adapt ideas and make compromises when implementing a new method. In this section, we touch upon aspects we needed to consider to ensure a reliable implementation of the framework proposed in this paper.

**Time as a measure of duration.** A heuristic schedule controls two general aspects: The order and the duration for which the different heuristics are executed. Even though it might seem intuitive to use time to control the duration of a heuristic run, we use a suitable proxy measure for every class of heuristics instead, as discussed in Section 3. There are two main problems that hinder us from directly controlling time. First, time is generally not stable enough to use for decision-making within an optimization solver. To make it somewhat reliable, we would need to solve instances exclusively on a single machine at a time. Hence, it would not be possible to run instances in parallel which would cause the solving process to be very expensive in practice. The second, even more important problem is the following. Since the behavior of heuristics significantly depends on different parameters, allowing the heuristic to run for a longer time does not necessarily translate to a increase in success probability if crucial parameters are set to be very restrictive by default. That is why we use a suitable proxy measure for time instead of time itself.

**Limitations of the shadow mode.** To make sure we obtain data that is as independent as possible, the heuristics run in shadow mode during data collection. This setting aims to ensure that the heuristics only run in the background and do not report anything back to the solver. However, it is not possible to hide *all* of the heuristic's actions from SCIP. Since SCIP is not designed to have heuristics running in the background, it is almost impossible to locate and adjust the lines of code that influence the solving process globally without limiting the heuristic's behavior too much. For instance, one way of hiding all actions of diving heuristics would be turning off propagation while diving. Since this would influence the performance of the heuristics considerably, the resulting data would not represent how the heuristics behave in practice. That is why we settled with a shadow mode that hides most (and the most influential) of the heuristic's activities from SCIP.

## F   Complete Results

In the following, we present more detailed results. Table 3 shows the complete results of the transfer learning experiments introduced in Section 6.2. Table 4 gives an overview of the performance of the different schedules used in the transfer learning experiments over all test sets.

| test
train | [150,160] | [200,210] | [250,260] | [300,310] | [350,360] | [400,410] | [450,460] | [500,510] | [550,560] |
|---|---|---|---|---|---|---|---|---|---|
| [150,160] | $0.89 \pm 0.23$ | $0.76 \pm 0.22$ | $0.87 \pm 0.37$ | $0.95 \pm 0.40$ | $0.87 \pm 0.28$ | $0.86 \pm 0.23$ | $0.78 \pm 0.24$ | $0.80 \pm 0.25$ | $0.65 \pm 0.24$ |
| [200,210] | $0.94 \pm 0.28$ | $0.75 \pm 0.25$ | $0.82 \pm 0.30$ | $0.91 \pm 0.34$ | $0.93 \pm 0.23$ | $0.90 \pm 0.28$ | $0.83 \pm 0.22$ | $0.79 \pm 0.20$ | $0.66 \pm 0.20$ |
| [250,260] | $0.89 \pm 0.28$ | $0.69 \pm 0.23$ | $0.81 \pm 0.34$ | $0.94 \pm 0.40$ | $0.92 \pm 0.23$ | $0.96 \pm 0.39$ | $0.81 \pm 0.24$ | $0.76 \pm 0.22$ | $0.66 \pm 0.20$ |
| [300,310] | $0.87 \pm 0.25$ | $0.71 \pm 0.26$ | $0.83 \pm 0.36$ | $0.97 \pm 0.39$ | $0.92 \pm 0.28$ | $0.90 \pm 0.35$ | $0.81 \pm 0.24$ | $0.75 \pm 0.24$ | $0.61 \pm 0.24$ |
| [350,360] | $0.84 \pm 0.24$ | $0.70 \pm 0.23$ | $0.82 \pm 0.36$ | $0.91 \pm 0.37$ | $0.81 \pm 0.26$ | $0.86 \pm 0.31$ | $0.80 \pm 0.21$ | $0.75 \pm 0.19$ | $0.59 \pm 0.20$ |
| [400,410] | $0.90 \pm 0.27$ | $0.70 \pm 0.23$ | $0.83 \pm 0.36$ | $0.88 \pm 0.32$ | $0.77 \pm 0.23$ | $0.88 \pm 0.30$ | $0.81 \pm 0.21$ | $0.74 \pm 0.20$ | $0.58 \pm 0.20$ |
| [450,460] | $0.89 \pm 0.25$ | $0.70 \pm 0.23$ | $0.83 \pm 0.36$ | $0.88 \pm 0.32$ | $0.77 \pm 0.23$ | $0.88 \pm 0.30$ | $0.81 \pm 0.21$ | $0.74 \pm 0.20$ | $0.58 \pm 0.20$ |
| [500,510] | $0.89 \pm 0.26$ | $0.72 \pm 0.22$ | $0.84 \pm 0.30$ | $0.99 \pm 0.42$ | $0.92 \pm 0.24$ | $0.95 \pm 0.46$ | $0.81 \pm 0.23$ | $0.80 \pm 0.25$ | $0.61 \pm 0.20$ |
| [550,560] | $0.88 \pm 0.26$ | $0.72 \pm 0.24$ | $0.89 \pm 0.42$ | $0.95 \pm 0.37$ | $0.86 \pm 0.27$ | $0.90 \pm 0.28$ | $0.81 \pm 0.20$ | $0.78 \pm 0.23$ | $0.63 \pm 0.21$ |
| SCIP_TUNED | $0.89 \pm 0.28$ | $0.77 \pm 0.23$ | $0.99 \pm 0.31$ | $1.08 \pm 0.45$ | $1.05 \pm 0.28$ | $1.03 \pm 0.38$ | $0.94 \pm 0.23$ | $0.91 \pm 0.28$ | $0.76 \pm 0.25$ |

Table 3: Average relative primal integral (mean $\pm$ std.) of schedules (with diving heuristics) w.r.t. default SCIP over GISP instances derived from random graphs. The first nine rows correspond to schedules that were trained on instances of size [L,U]

| SCHEDULE | better primal integral | better primal bound | incumbents found by schedule heuristics |
|---|---|---|---|
| [150,160] | 0.69 | 0.70 | 0.61 |
| [200,210] | 0.69 | 0.65 | 0.58 |
| [250,260] | 0.68 | 0.55 | 0.49 |
| [300,310] | 0.72 | 0.58 | 0.50 |
| [350,360] | 0.76 | 0.62 | 0.51 |
| [400,410] | 0.75 | 0.61 | 0.52 |
| [450,460] | 0.75 | 0.61 | 0.52 |
| [500,510] | 0.68 | 0.58 | 0.50 |
| [550,560] | 0.70 | 0.59 | 0.50 |
| SCIP_TUNED | - | - | 0.44 |
| DEFAULT | - | - | 0.33 |

Table 4: Fraction of instances for which the learned schedules (with diving heuristics) have a better primal integral/bound at termination w.r.t. SCIP_TUNED (second and third column) and fraction of incumbents that where found by heuristics in the schedule (fourth column). Only instances that were not solved to optimality by both SCIP_TUNED and the schedule are considered in the third column.