# OpenReview forum: "Learning to Schedule Heuristics in Branch and Bound"
_NeurIPS.cc/2021/Conference — NeurIPS 2021 Poster_

### Official Review · Reviewer_D5iZ · 2021-07-12

**Rating:** 7
**Confidence:** 4

**Summary:**

**The problem:** This paper studies a machine learning approach to scheduling primal heuristics in branch-and-bound. Primal heuristics are used throughout the branch-and-bound tree to find feasible solutions to the integer program. These feasible solutions help speed up the time it takes branch-and-bound to find the optimal solution. There are a few challenges that arise when using heuristics. First, there are many different heuristics one could employ with varying performance across application domains. Second, there is an inherent time-utility tradeoff: the more time you allocate to a heuristic, the more likely it is that it will find a new incumbent solution, but branch-and-bound’s total runtime may increase. Given a set of heuristics $H$, this paper uses a training set of integer programs from the particular application domain at hand to learn a schedule $\langle(h_1, t_1), …, (h_k, t_k)\rangle$ of heuristics $h_i \in H$ and budgets $t_i > 0$ such that at each node, branch-and-bound will run each heuristic hi in the schedule in order, allocating ti iterations to the heuristic.

**The approach:** First, we describe the data collection process. We begin with a set of integer programs from the application domain at hand. Then, while solving each integer program, at each node $N$, we see for each heuristic $h \in H$ how many iterations $t$ it takes to find a feasible solution. The training set consists of all such tuples $(N, h, t)$. The authors show that finding the optimal schedule over the training set can be formulated as a mixed-integer quadratic program, but solving it is not generally tractable. Therefore they propose a greedy heuristic algorithm, which at each step (essentially) adds the heuristic-budget pair $(h, t)$ that maximizes the marginal increase in the number of instances solved divided by $t$ (which is the cost of including $(h, t)$ in the schedule). (Their approach is a bit more nuanced than this, but this is the high-level idea.)

The results: The primary metric that the authors optimize is the “primal integral,” which can be interpreted as a normalized average of the incumbent value over time. We want to find a good incumbent as quickly as possible, so we want the primal integral to be small. They provide experiments for two different classes of integer programs and show that their approach generally improves over a few baselines including default SCIP, a carefully tuned implementation of SCIP, and schedules optimized using SMAC [Hutter et al., LION’11].

**Limitations And Societal Impact:**

I appreciated the discussion of limitations in the final section.

**Main Review:**

Overall, I thought that this paper’s approach to learning heuristic schedules was insightful and well thought-out; I appreciated the example illustrating why Streeter’s greedy algorithm would not work out-of-the-box. I also appreciated the paper’s scalable approach to data collection, since this is a big challenge in research on machine learning for integer programming. Moreover, I thought that the paper was written well overall, though I have a few writing suggestions later in this review.

In the experiments section, I would have appreciated a bit more clarity on how the results ultimately translate to branch-and-bound’s overall performance. In Table 2, it seems like if the total number of heuristic calls is less than the baselines and the total time spent on heuristics is also less, then B&B should run faster overall, but it would be helpful if the authors could verify this and connect those dots.

I’d also be interested in some discussion of whether it would be possible to implement these ideas using CPLEX callback functions, for example, or if there are some big hurdles to doing so. Improving SCIP itself seems quite worthwhile, but I suppose that an integration with CPLEX or Gurobi would be more practically relevant.

In terms of writing, I have some detailed comments below, but a higher-level comment is that a fair amount of space was devoted to the MIQP formulation of the problem, but I’m not sure if it was worth the space since it’s not the path the authors end up pursuing. It helped motivate the example in lines 187-192, but maybe the example can be motivated even without the MIQP formulation.

Detailed Comments:

- Line 143: I prefer “scalable” over “computationally efficient”, since you do still have to run B&B, so I’m not sure that data collection can actually be considered computationally efficient.
- Line 195: There seems to be a noun missing from the sentence “That is, we allow…”
- Lines 193-197: With this notation, shouldn’t $h_{j-1}$ be $g_{j-1}$? The same comment holds for line 205.
- Line 200: “lead” -> “led”
- Line 297: “heurisitcs” -> “heuristics”
- Lines 307-308: I see what the authors means when they say that only scheduling diving heuristics makes it easier for SMAC to terminate, but does this also make it harder for SMAC to compete?
- Lines 329-333: Ideally, it would be awesome to run Khalil et al.’s approach on SCIP 7.0 in order to compare it to this paper’s approach.
- Lines 334-336: These lines say that “the schedule setting dominates both default SCIP and SCIP_TUNED in all metrics,” but in the third row, don’t these two baselines dominate?
- Table 2: I was a bit confused at first by the “best incumbent row” --- is the GSIP objective maximization and the FCMNF objective minimization? If so, it would be helpful to clarify. Also, why isn’t anything highlighted in rows 4 and 5?

**Time Spent Reviewing:**

5

---

> ### Author Response · Authors · 2021-08-10
> **Response to Reviewer D5iZ**
>
> We thank the reviewer for carefully reading our paper, their detailed comments and the positive remarks regarding the overall presentation of our manuscript. We especially welcome the reviewer's assessment that “this paper’s approach to learning heuristic schedules was insightful and well thought-out“. Several valuable suggestions have been given, which we address individually:
>
> 1. Since most of our instances hit the time limit of two hours before finding an optimal solution, we need to look at other metrics than the total solving time to analyze the impact on B&B’s overall performance. Table 2 shows that the schedule settings terminated with a better primal bound as well as a smaller primal-dual gap for both GISP and SMAC suggesting an improvement in the overall performance when solving these problems to optimality. This behavior was also observed on the three smallest cross-validation test sets ([150,160], [200,210], [250,260]) which further supports this assumption.
>
> 2. Even though SCIP is a state-of-the-art academic solver, we agree with the reviewer that it would also be interesting to integrate our approach into other solvers like CPLEX and Gurobi. Since our method does not use any specific features of SCIP, it is possible to use any other solver given that one has access to its source code. To learn a heuristic schedule, we need to collect data about the behaviour of the different heuristics. Since this information is not typically given by a solver, data collection is only possible by manipulating the solver's source code to implement the shadow mode described in our paper. Hence, we rely on open-source solvers, which unfortunately is not the case for CPLEX and Gurobi.
>
> 3. We decided to include the MIQP formulation since it helps to formally define the heuristic scheduling problem first, before introducing an efficient approach to solving it. This way, we avoid giving a hand-wavy description of the learning task, but instead characterising it completely. Furthermore, we think that in future work we will be able to derive approximation guarantees for the greedy approach similar to the ones derived for Streeter’s problem. Hence, having the learning task formally defined is the first step to deriving learning theoretical results.
>
> 4. We compare the schedules trained with SMAC to greedy schedules that only include diving heuristics. That is why the comparison is fair.
>
> 5. We are currently working to implement the method of Khalil et al. (2017) within the Ecole framework for learning in SCIP (https://www.ecole.ai/), which we believe is the best way to compare transparently. We do believe, however, that the overhead of computing features at each node of the search tree, coupled with the lack of control over the number of iterations of each heuristic, make Khalil et al.’s method less suitable in practice.
>
> 6. Lines 334-336: We agree with the reviewer that the formulation is misleading and needs to be corrected. We have done so in the revised manuscript. Since the problems in this test set are not solved to optimality, it is important to also compare the objective function values of the best incumbent found when looking at the time necessary to find the best incumbent. As we can see, SCHEDULE finds a better incumbent in both cases, which explains the increase in the time necessary to find it.
>
> 7. Table 2: As remarked by the reviewer, GISP is a maximisation whereas FCMNF is a minimisation problem. We have added further clarification in the revised manuscript. To address the reviewer’s question about the missing highlights in row 4 and 5, we would like to note that highlighted entries should indicate best performance of one setting w.r.t. a performance metric. Rows 4 and 5 show important statistics rather than metrics that indicate better or worse primal performance (e.g., spending less/more time in heuristics can have a good or a bad impact on the overall primal performance). That is why we omitted highlighting entries in those rows.

---

> > ### Comment · Reviewer_D5iZ · 2021-09-08
> > **Thank you.**
> >
> > Thank you for the detailed comments! I'll maintain my evaluation that the paper should be accepted.

---

### Official Review · Reviewer_JnKT · 2021-07-13

**Rating:** 7
**Confidence:** 4

**Summary:**

The paper studies the scheduling of multiple heuristics inside of a mixed-integer programming solver. The authors formulate the learning problem exactly, and then present a heuristic method with greater scalability. They then present computational results validating the efficacy of their approach.

**Main Review:**

The paper is well written. It builds off papers from the literature, but makes its case for its originality. The significance of the work is bolstered by the computational results. However, I have a concern about the baseline used for comparison. More fundamentally, I am not sure that the result paper matches exactly with its stated goal.

* P1: Good primal solutions are important for another reason: they can help reduce search tree size, which can lead to faster convergence when proving optimality.
* P2: What is the "success rate"? Can the authors provide a reference and/or a brief description?
* L73: The description of how the authors go from node to UUID in such a way that this is meaningful across instances is very vague, and I would imagine that this is actually very important from a practical perspective. How does this work for problem instances with varying number of variables? Is it robust to shuffling variable indices? Etc.
* L135: Weighing uniformly across all nodes seems like it may be suboptimal. Intuitively, I would imagine that weighing nodes near the top of the tree (e.g. the root) over ones deeper in the tree would lead to better schedules inside a MIP solver. Have the authors considered this?
* L161: Point 1 is, frankly, not true without softening or qualification. A cheeky "counterexample" would be to point out that each MIP can trivially be transformed into a decision problem (constrain the objective function to be equal to the optimal cost). Additionally, many solvers for decision problems (e.g. SAT) are also tree-based, with multiple opportunities to run heuristics on different subproblems.
* L194: Why only consider the last heuristic, rather than all preceding heuristics?
* L297: "heurisitcs"
* The usage of SCIP with default settings is a bit of a strawman. The solver are configured to prove optimality quickly, while the authors only are concerned with primal solutions. As such, SCIP will spend significant times improving the dual bound through cuts, presolve, etc, which renders wall time comparisons unfair. A more fair comparison would configure the solver to focus on the primal bound, using the "emphasis" parameter.
* The previous comment dovetails into my fundamental concern about the framing of the paper. The authors present the work as a method for improving primal heuristics inside a MIP solver, but the contributions are instead framed more as a metaheuristic for producing primal solutions for MIPs. The latter is a fine goal, but fundamentally different from the former, as a MIP solver by default will spend considerable time and effort on the dual bound. To argue for the former, the authors should really present comparisons on the primal-dual gap integral, which take into account the significant effort SCIP spends on the dual side.

**Time Spent Reviewing:**

1.5 hours

---

> ### Author Response · Authors · 2021-08-10
> **Response to Reviewer JnKT**
>
> We thank the reviewer for a careful reading of our manuscript, their detailed comments and the positive remarks regarding the overall presentation of our manuscript. In the following, we address the reviewers detailed comments point by point:
>
> 1. P2: The (solution) success rate of a heuristic is the number of times a solution was found by the heuristic divided by the total number of calls to that heuristic. We clarified this in the revised manuscript.
>
> 2. L73: In a MIP solver, the nodes of the B&B tree are often numbered to differentiate nodes. We use this node index to uniquely identify a node such that we can easily track which heuristics were executed at which nodes during the data collection phase. Since the goal of our framework is to derive a single schedule for a class of training instances, we do not use features that characterise different nodes, but treat all nodes equally. Neither the number of variables nor the variable indices play a role for this.
>
> 3. L135: Thank you for this interesting idea. We have not tested the weighting scheme which you propose, but will consider doing so.
>
> 4. L161: When solving a decision problem, there exists a clear answer; either „yes“ or „no“. This is not the case for MIPs. Since each feasible solution can be potentially optimal, it is generally not possible to say beforehand if a solution found by a heuristic can be dismissed or not. That is why we need to approach the problem of optimizing the use of heuristics differently for MIPs than for decision problems: Instead of dismissing heuristics that do not find the optimal solution right away, we need to prioritize heuristics that find (good) solutions often. As the reviewer remarked, MIPs can also be stated as a decision problem given that the optimal solution is known beforehand but this is not how a MIP is solved in practice. In general, knowing the optimal solution value beforehand does not speed up the MIP solver very much because proving optimality is translated into proving the non-existence of a better solution and that remains hard.
>
> 5. L194: Allowing each heuristic to be added only once is not our decision: Heuristic management in a MIP solver is designed to execute every heuristic at most once whenever the solver decides to run heuristics. Hence, to obtain a schedule that is compatible with the solver, we adopted this restriction. That is why we only consider the last heuristic rather than all preceding heuristics. The latter would produce a schedule that might contain a heuristic multiple times, whereas the former only considers the option to run the last heuristic for longer.
>
> 6. First, we would like to note that when testing the trained schedules, we only change the respective heuristics settings and have no influence over how SCIP decides to run other components like presolving, cuts, etc. Hence, the schedule settings we consider in our experiments have the same time allocated to improving the dual bound as SCIP default. Even when considering not only the primal improvement (as the primal integral does), but also the dual improvement we see that the scheduling approach outperforms both SCIP and SCIP_TUNED: In Table 2, we see that SCHEDULE reduces the primal-dual gap at termination, as well as the primal-dual integral for both GISP and FCMNF over the other settings. As the reviewer mentioned, SCIP has different „emphasis“ settings that are more tailored to proving feasibility instead of optimality of the problem. We observed that those settings perform not only significantly worse than the schedules, but also worse than SCIP’s default settings.
>
> 7. Our framework is not supposed to choose a suitable heuristic to apply at a given moment, but controls the application of heuristics on the general heuristic management level of a MIP solver. That is why we think our method aims to improve primal heuristics inside a MIP solver. As mentioned above, the schedule only influences the behaviour of primal heuristics, hence the MIP solver still decides freely how much time to allocate to other solving components (e.g., components that aim to improve the dual bound). We agree with the reviewer that looking at the primal integral alone is not enough to get a complete picture. That is why we consider multiple other metrics in Table 2. Besides reducing the primal-dual integral (row „Primal-dual Integral“) as well as the primal-dual gap (Row „Gap"), the trained schedule finds more incumbents (Row „Number of Incumbents found“) and increases the probability that a heuristic finds a new incumbent („Incumbent Success Rate“) while reducing the overall time spent in heuristics (Row „Total heuristic time“). The latter shows that primal performance is not improved by simply allocating more time to heuristics and ignoring the dual side of the problem, but intelligently applying primal heuristics.

---

> > ### Comment · Reviewer_JnKT · 2021-08-31
> > **Response**
> >
> > I thank the authors for their response to my questions about the computations. I am happy with the responses in that area, and will increase my score correspondingly. However, I still have some concerns about the presentation and framing of the paper.
> >
> > >  L73: In a MIP solver, the nodes of the B&B tree are often numbered to differentiate nodes. We use this node index to uniquely identify a node such that we can easily track which heuristics were executed at which nodes during the data collection phase. Since the goal of our framework is to derive a single schedule for a class of training instances, we do not use features that characterise different nodes, but treat all nodes equally. Neither the number of variables nor the variable indices play a role for this.
> >
> > I see, so the UUID is the number of nodes the solver has processed before a given node? And not some (more complex) ID that considers e.g. the branching decisions used to reach that node. Considering that how this done is opaque and depends intricately on many details of how the solver implements the tree search, I ask that the authors clarify the description in the paper.
> >
> > > L161: When solving a decision problem, there exists a clear answer; either „yes“ or „no“. This is not the case for MIPs. Since each feasible solution can be potentially optimal, it is generally not possible to say beforehand if a solution found by a heuristic can be dismissed or not. That is why we need to approach the problem of optimizing the use of heuristics differently for MIPs than for decision problems: Instead of dismissing heuristics that do not find the optimal solution right away, we need to prioritize heuristics that find (good) solutions often. As the reviewer remarked, MIPs can also be stated as a decision problem given that the optimal solution is known beforehand but this is not how a MIP is solved in practice. In general, knowing the optimal solution value beforehand does not speed up the MIP solver very much because proving optimality is translated into proving the non-existence of a better solution and that remains hard.
> >
> > I am not sure this is responsive to my comment. I pointed out that MIPs are not more difficult than decision problems in a theoretical sense. I guess the authors could argue that this is true in a practical sense, but this is a very strong claim — “decision problems” is an incredibly broad class!--and calls for some marshalling of an argument beyond a simple assertion in ambiguous terms. Hence my request for softening and/or qualification.
> >
> > > L194: Allowing each heuristic to be added only once is not our decision: Heuristic management in a MIP solver is designed to execute every heuristic at most once whenever the solver decides to run heuristics. Hence, to obtain a schedule that is compatible with the solver, we adopted this restriction. That is why we only consider the last heuristic rather than all preceding heuristics. The latter would produce a schedule that might contain a heuristic multiple times, whereas the former only considers the option to run the last heuristic for longer.
> >
> > Is this claim about heuristic management generalizable across the range of modern MIP solvers (e.g. Gurobi, CPLEX, Xpress, Cbc, etc.), or is it particular to the API exposed by SCIP? Is this property of how SCIP works documented as part of the API, or is it an implementation detail (that may change at any time)? (A quick skim of the SCIP documentation didn’t turn up anything, though I very well may have missed it). As one example of why a solver may decide _not_ to follow this schema, it seems plausible that a solver may want to run a very quick local search heuristic on top of a solution just produced by a much more expensive method.
> >
> > More generally, this design decision is not set in stone: you could write a “meta heuristic” that wraps around a number of standard heuristics with its own logic about when they are run and how. Or, if you can modify the solver internals, you could directly configure the heuristic management logic to behave differently. The first is what you would likely have to do anyway if you wanted to implement this in e.g. Gurobi using its heuristic callback API, and the second is currently possible in SCIP as it is “open” (modulo licensing restrictions).
> >
> > Choosing to structure the proposed algorithm around the particulars of SCIP’s API is perfectly defensible, but I view it primarily as a modeling  decision, and should be presented and defended as such.
> >
> > > First, we would like to note that when testing the trained schedules, we only change the respective heuristics settings and have no influence over how SCIP decides to run other components like presolving, cuts, etc. Hence, the schedule settings we consider in our experiments have the same time allocated to improving the dual bound as SCIP default.
> >
> > In the presence of a time limit, is this really true? Based on the authors model, if I ordered the heuristics so that very fast heuristics produced very bad solutions early in the heuristic loop, this would seemingly allow more time in the budget for work on the dual bound (e.g. by enumerating more nodes).

---

### Official Review · Reviewer_8aHm · 2021-07-15

**Rating:** 7
**Confidence:** 2

**Summary:**

The paper proposes a method to learn a heuristic schedule in branch and bound solvers, e.g. for mixed integer programming.
By learning from collected data an efficient heuristic can be selected more efficiently than from hand-crafted expert rules, so the claim.
Experimental results show that the proposed method is indeed effective and improves the average primal integral.

**Limitations And Societal Impact:**

Limitations and societal impact are sufficiently addressed.

**Main Review:**

The work is well-motivated and clearly explained.
What I personally like is the approach to data collection by executing multiple heuristics at the same time to gather more information in a single run of the optimizer.
The trained model is then embedded in the solver at test time to propose a heuristic schedule.
While this area has drawn some attention, the related work section of this paper is relatively small but also highly focused on one small subarea of the field, which is okay.

What I find a bit particular is the selection of experimental problems, which appear to not be very common problems (as far as I know). I'm wondering why especially these problems were selected and why not more prominent problems were chosen to make comparisons among the literature easier.

The evaluation focuses on the primal integral were strong reductions are possible, but which is not the most interesting metric for potential users, although otherwise a good performance indicator.
The shown reduction however also translates to improved solving times, although the tuned SCIP still is close or outperforms on the time to best incumbent. For FCMNF even the default configuration has the best performance for time to best incumbent, which is surprising. Is there an intuition why this is the case?

Finally, how does the method generalize? Can collected information from one problem be transferred to other problems or is it strictly necessary to collect a new dataset each time?

In general, I think this is solid work and the integration of ML into exact (branch-and-bound) solvers is of high interest, but I'm not entirely convinced that we are there yet to actually make strong improvements over the large body of experience and knowledge in solver design and tuning.
Nevertheless, I do not see a strong reason why this paper should not be accepted and contribute to the body of work leading to stronger solvers.

**Time Spent Reviewing:**

3

---

> ### Author Response · Authors · 2021-08-10
> **Response to Reviewer 8aHm**
>
> We thank the reviewer for their thorough review of our manuscript and the positive remarks regarding the overall presentation of the manuscript and our data collection methodology. In the following, we address the reviewer’s detailed comments point by point.
>
> 1. Since our heuristic scheduling method aims to improve the primal performance of a MIP solver,  problems that are challenging on the primal side are best suitable to investigate the method’s performance. Both GISP and FCMNF are observed to be hard on the primal side and thus preferable over more common benchmark problems.
>
> 2. The primal integral is a commonly used metric for analysing primal performance. For instance, the Machine Learning for Combinatorial Optimization NeurIPS 2021 competition, whose  aim is to improve solvers using ML, uses the primal integral as the only performance indicator in the primal task. Nonetheless, as with all performance metrics, the primal integral comes with advantages and disadvantages. That is why we also consider other metrics, like „the time to best incumbent“, to give a complete picture of the computational results. As the reviewer correctly mentioned, Table 2 shows that the setting SCIP_TUNED (SCIP) needs less time to find the best incumbent than the schedules on GISP (FCMNF). Since the problems in these test sets are not solved to optimality, it is important to also compare the objective function values of the best incumbent found by the settings. As we can see, SCHEDULE finds a better incumbent in both cases (note that GISP is a maximisation and FCMNF a minimisation problem), which explains the increase in time necessary to find it.
>
> 3. It is hard to use information about the behavior of heuristics for one problem to improve performance on another problem, since a heuristic’s behaviour is highly problem-dependent (see Figure 1). Our method generalises in the sense that even when the model is trained on small problems, it significantly improves primal performance on much bigger problem instances (see Table 1). This is a setting that is especially important for real-world applications.

---

> > ### Comment · Reviewer_8aHm · 2021-08-18
> > **Response to Authors**
> >
> > Thank you for your response and the clarification to my questions.
> > Given this and the information from the other responses, I am increasing my score to support the paper.

---

### Official Review · Reviewer_Nehp · 2021-07-16

**Rating:** 6
**Confidence:** 4

**Summary:**

The authors propose a data-driven approach for learning the optimal scheduling heuristics to solve the MILP problems. They formalize the scheduling problem mathematically and suggest an efficient data collection strategy. In experiments, they show the advantage of their algorithms by comparing them with state-of-the-art models over several different datasets.

**Ethical Concerns:**

No.

**Limitations And Societal Impact:**

Yes. The authors adequately addressed the limitations and potential negative societal impact of their work

**Main Review:**

The authors propose a data-driven approach for learning the optimal scheduling heuristics to solve the MILP problems. They formalize the scheduling problem mathematically and suggest an efficient data collection strategy. In experiments, they show the advantage of their algorithms by comparing them with state-of-the-art models over several different datasets.

Strengths:

The authors propose an interesting and important perspective for solving the MILP problems. They observe that having a better solution early on would lead to faster minimization of the primal gap. Most  MILP problems require a large amount of time for tuning the scheduling parameters. As a consequence, they suggest that that the scheduling of MILP can be formalized into a machine learning problem.

Points of improvement:

-	table 1 mainly shows that the proposed algorithm guides the heuristics in an early timestep, so the primal integral value can be improved. I’m curious about the effect of reducing the whole time for solving MILP solvers. In table 2, the authors present some numbers regarding the “time to first incumbent” and “time to best incumbent”. I’m looking forward to a more detailed table/figure about that.

-	Even though the author claims that the algorithm is not tied to the single solver. We would like to see if they can run a few more experiments on some other solvers to justify this claim.

-	Line 338 needs to be deleted. Some references are not correct, for example, in line 24, line 78 and etc.


**Time Spent Reviewing:**

3 hours

---

> ### Author Response · Authors · 2021-08-10
> **Response to Reviewer Nehp**
>
> We thank the reviewer for a careful reading of our manuscript, their detailed comments and the kind characterisation of our heuristic scheduling approach as „an interesting and important perspective for solving MILP problems“. We will now address the reviewers listed questions point-by-point.
>
> 1. Since most of our instances hit the time limit of two hours before finding an optimal solution, we need to look at other metrics than the total solving time to analyze the impact on B&B’s overall performance. Table 2 shows that the schedule settings terminated with a better primal bound as well as a smaller primal-dual gap for both GISP and SMAC suggesting an improvement in the overall performance when solving these problems to optimality. This behavior was also observed on the three smallest cross-validation test sets ([150,160], [200,210], [250,260]) which further supports this assumption. We added a more detailed discussion to the revised version of our paper.
>
> 2. Even though SCIP is a state-of-the-art academic solver, we agree with the reviewer that it would be also interesting to integrate our approach into other solvers. Since our method does not use any specific features of SCIP, it is possible to use any other solver given that one has access to its source code. To learn a heuristic schedule, we first need to collect data and then run the greedy algorithm to obtain the schedule (this is done by a separate script). The latter will produce a schedule that translates to a settings file which can be given to any solver as an input. Hence, the only component that is tied to a solver is the data collection step. Since the necessary information is not typically given by a solver, data collection is only possible by manipulating the solver's source code to implement the shadow mode described in our paper. That is why we need to rely on open-source solvers.
>
> 3. Unfortunately, we are not sure why line 338 needs to be deleted and why the references in line 24 and 78 are wrong. We would appreciate it if the reviewer could elaborate on that more.

---

> > ### Comment · Reviewer_Nehp · 2021-08-20
> > **I have read the response from the authors and I maintain my score.**
> >
> > Overall, I find all reviewers think positively about this paper.

---

### Decision · Program_Chairs · 2021-09-28

**Decision:**

Accept (Poster)

**Comment:**

In combinatorial optimization, the design and selection of branching heuristics play a key role. Such heuristics are often carefully hand crafted and their performance is very much dependent  on the particular instances. In fact, the time performance of branching heuristics can vary exponentially  from instance to instance.  The paper proposes a data driven approach for learning  heuristic schedules  for exact  (i.e., with optimality guarantees) Mixed Integer Programming. They formulate the scheduling problem and propose an efficient data collection strategy. The schedule is problem/instance dependent and results show significant improvement with respect to the primal gap, a key measure used in exact MIPs. There was consensus about this paper and the reviewers carefully consider the authors' feedback.

**Consistency Experiment:**

NeurIPS has a long history of experimentation. In 2014, NeurIPS ran an experiment in which 10% of submissions were reviewed by two independent committees to quantify the randomness in the review process. This year, we repeated a variant of this experiment to see how the quality of the review process has changed over time.  This paper was part of the experiment and was therefore assigned to two committees (consisting of reviewers, an Area Chair, and a Senior Area Chair) that reached independent decisions.  If both committees made the same recommendation, this recommendation was followed. If a single committee recommended acceptance, the paper was accepted (with the exception of a few cases in which the other committee identified what we considered a fatal flaw, e.g., an error in a key result).

Both committees reached the same decision: **Accept (Poster)**

The other committee assigned to the paper recommended **Accept (Poster)**.  You can find the other set of reviews, along with any follow up discussion with the authors here:
https://openreview.net/forum?id=mvEhkIqn45_